# Perturbations of the Proteome and of Secreted Metabolites in Primary Astrocytes from the hSOD1(G93A) ALS Mouse Model

**DOI:** 10.3390/ijms22137028

**Published:** 2021-06-29

**Authors:** Roberto Stella, Raphael Severino Bonadio, Stefano Cagnin, Maria Lina Massimino, Alessandro Bertoli, Caterina Peggion

**Affiliations:** 1Department of Chemistry, Istituto Zooprofilattico Sperimentale delle Venezie, 35020 Legnaro, Italy; rstella@izsvenezie.it; 2Department of Biology and CRIBI Biotechnology Center, University of Padova, 35131 Padova, Italy; raphaelbonadio@gmail.com (R.S.B.); stefano.cagnin@unipd.it (S.C.); 3CIR-Myo Myology Center, University of Padova, 35131 Padova, Italy; 4CNR—Neuroscience Institute, 35131 Padova, Italy; marilina.massimino@gmail.com; 5Padova Neuroscience Center, University of Padova, 35131 Padova, Italy; 6Department of Biomedical Sciences, University of Padova, 35131 Padova, Italy

**Keywords:** spinal cord astrocytes, ALS, glutathione metabolism, proteolysis, metabolomics, proteomics, transcription factors

## Abstract

Amyotrophic lateral sclerosis (ALS) is a progressive neurodegenerative disease whose pathophysiology is largely unknown. Despite the fact that motor neuron (MN) death is recognized as the key event in ALS, astrocytes dysfunctionalities and neuroinflammation were demonstrated to accompany and probably even drive MN loss. Nevertheless, the mechanisms priming astrocyte failure and hyperactivation are still obscure. In this work, altered pathways and molecules in ALS astrocytes were unveiled by investigating the proteomic profile and the secreted metabolome of primary spinal cord astrocytes derived from transgenic ALS mouse model overexpressing the human (h)SOD1(G93A) protein in comparison with the transgenic counterpart expressing hSOD1(WT) protein. Here we show that ALS primary astrocytes are depleted of proteins—and of secreted metabolites—involved in glutathione metabolism and signaling. The observed increased activation of Nf-kB, Ebf1, and Plag1 transcription factors may account for the augmented expression of proteins involved in the proteolytic routes mediated by proteasome or endosome–lysosome systems. Moreover, hSOD1(G93A) primary astrocytes also display altered lipid metabolism. Our results provide novel insights into the altered molecular pathways that may underlie astrocyte dysfunctionalities and altered astrocyte–MN crosstalk in ALS, representing potential therapeutic targets to abrogate or slow down MN demise in disease pathogenesis.

## 1. Introduction

Amyotrophic lateral sclerosis (ALS) is a late-onset neurodegenerative disorder characterized by the selective loss of upper and lower motor neurons (MNs) in the spinal cord, brainstem, and cerebral cortex [1,2,3,4]. Progressive MN degeneration leads to paralysis and eventual death within 2–5 years from symptom onset with an estimated worldwide mortality of 30,000 people per year [5,6,7].

Although the majority of ALS cases occur sporadically (sALS), nearly 10% of ALS patients have inherited familial ALS (fALS). To date, more than 20 genes related to fALS were identified. Among such genes, mutations in superoxide dismutase 1 (SOD1), transactive response DNA binding-43 kDa (TDP-43), fused in sarcoma/translocated in sarcoma (FUS/TLS), and the most frequent intronic hexanucleotide GGGGCC repeat expansion in the C9orf72 gene (accounting for the 40–50% of fALS and 7% of sALS) are responsible for approximately the 70% of all fALS monogenic cases [8,9].

fALS and sALS are phenotypically indistinguishable, whereas each of the above ALS-related genes play different functions in a variety of cell pathways, hampering a unifying view of the pathogenic mechanisms. Advancements towards understanding the pathogenic pathways in ALS over the past years have unveiled a pleiotropy of deregulated processes. Among these, oxidative stress, mitochondrial dysfunction, ER stress, glutamate excitotoxicity, protein misfolding/aggregation, defects in axonal transport, aberrant RNA metabolism, and glial-mediated neuroinflammation suggested multifactorial pathomechanisms in ALS [9,10,11].

Different aspects of ALS pathogenesis have been discovered thanks to studies using rodent models of inherited forms of ALS. The first and currently most used mouse model is that overexpressing the G93A missense mutant of human (h) superoxide dismutase 1 (SOD1) [12]. These transgenic (Tg) mice recapitulate many of the pathological hallmarks of ALS patients, including MN degeneration, astrogliosis, and the accumulation of SOD1-containing inclusions in MNs and glial cells [13,14].

Such animal models contributed to define ALS as a “multi-systemic” disease, not only affecting MNs, but rather causing general alterations in the structural, physiological, and metabolic pathways in different cell types (MNs, glia and muscle cells) that synergistically exacerbate disease progression. However, the precise mechanisms by which specific cell types contribute to the disease remain unknown. In this respect, it has been proposed that the disruption of glial cell–MN communication contributes to MN injury in ALS [9,15,16,17].

In this framework, astrocytes participate in many physiological processes (e.g., the delivery of antioxidant molecules or the modulation of synaptic transmission) ensuring neuronal health, and any perturbation of these key functions may lead to neuronal impairment. Intriguingly, several lines of evidence suggest that astrocytes undergo profound alterations in ALS, including modified morphology and protein expression patterns [18,19,20,21], alterations of intracellular or secreted metabolome [22,23,24], Ca^2+^ dyshomeostasis, mitochondrial functional deficiency, and increased oxidative stress [25,26,27,28]. Moreover, ALS astrocytes have been suggested to contribute to MN degeneration by altered secretion of trophic or neurotoxic factors [15,17,29,30,31,32].

Considering the above notions, we accomplished a thorough analysis of the secreted metabolome and of the cellular proteome of primary astrocytes from the spinal cord of newborn Tg mice expressing the ALS-related hSOD1(G93A) or the non-pathogenic wild-type (WT) hSOD1 using non-targeted metabolomics and proteomics approaches.

In this research, we showed that hSOD1(G93A) astrocytes present an altered secreted metabolome and profound alterations in the expression of proteins involved in proteostasis and glutathione (GSH) metabolism. Targeted analysis indicated that enhanced activity of the transcription factors (TFs) NF-kB, Ebf1, and Plag1 may account for altered protein expression in fALS astrocytes.

## 2. Results

### 2.1. Conditioned Medium Analysis by Ultra-High Performance Liquid Chromatography-High Resolution Mass Spectrometry (UHPLC-HRMS)

The metabolomic profiles of the conditioned medium (CM) of primary cultures of spinal cord astrocytes obtained from hSOD1(WT) and hSOD1(G93A) mice (15 samples for each genotype) were acquired using UHPLC-HRMS. Metabolites were separated by hydrophilic interaction liquid chromatography (HILIC) and analysed by HRMS both in positive and negative electrospray ionization polarity.

The two UHPLC-HRMS datasets consisted of 2050 and 1550 ions for positive and negative ionization modes, respectively. After data filtration (see Materials and Methods), the two datasets were reduced to 1176 and 779 molecules in the positive and negative ionization mode, respectively.

An unsupervised multivariate principal component analysis (PCA) of quality control (QC) (see Materials and Methods) and CM sample datasets was used to check the performance of the entire analysis. PCA score plots showed that both in negative (Appendix A, panel A) and positive (Appendix A, panel B) ionization modes, QC samples are well clustered together and separated from CM samples, suggesting the absence of strong deviation during data acquisition.

Then, we applied PCA to give a general overview of the bidimensional distribution of the 30 CM samples and to display sample grouping. The PCA score plot based on the LC-HRMS data acquired in the negative ionization mode displayed a clustered pattern for the 30 samples, segregating them into two clearly distinct groups according to the hSOD1 genotype and revealing a distinct global metabolomics signature for each group (Figure 1A).

Conversely, data acquired in the positive ionization mode did not display an appreciable separation between the hSOD1(G93A) and hSOD1(WT) samples, showing overlapping groups and more dispersed samples along the two PCs in the PCA score plot (Appendix A). Since this result suggested a major variance across hSOD1 genotypes, we decided to exclude such dataset from subsequent investigations.

Among the 779 compounds detected and relatively quantified in the negative ionization mode, 54 metabolites were found to be significantly affected by the hSOD1(G93A) genotype with respect to the hSOD1(WT) astrocytes (Figure 1B).

Of these 54 significantly altered metabolites, only 19 were annotated based on the measured *m*/*z* value of the precursor ion, the isotopic pattern, and the MS/MS fragmentation spectra, when present, by searching in the METLIN and the Kyoto Encyclopedia of Genes and Genomes (KEGG) databases (Appendix A). Using the KEGG pathway database, we investigated which metabolic pathways were principally altered. Interestingly, annotated and differentially expressed metabolites were involved in fatty acid biosynthesis and amino acid metabolism (Figure 1C). In particular, we found a significant increase of seven short-, medium- and long-chain free fatty acids (FFA) and a halved abundance of phosphatidic acid in CM derived from hSOD1(G93A) astrocytes compared to hSOD1(WT) cells. Moreover, a reduced amount of γ-glutamil-tyrosine (involved in the metabolism of GSH) and cysteine-S-sulfate were detected in CMs from hSOD1(G93A) astrocytes (Appendix A).

### 2.2. Proteomics Analysis of Primary Cultured Spinal Astrocytes

To better define the alterations in hSOD1(G93A) astrocytes, we also performed a comparative profiling of the cellular proteome of primary astrocytes (three different cell cultures for each hSOD1 genotype) using shot-gun proteomics based on UHPLC coupled with high resolution tandem mass spectrometry (HRMS/MS) and the relative quantification of proteins by the tandem mass tags (TMT) approach. The MS proteomics data have been deposited via the PRIDE partner repository [33] with the dataset identifier PXD026576.

Proteomics data were evaluated using the score plot of a PCA analysis of the obtained relative quantification values. As reported in Figure 2A, such analysis provided a clustered pattern for the six samples, separating them into two distinct groups according to the hSOD1 genotype. Within both the hSOD1(WT) and hSOD1(G93A) groups, a total of 640 proteins were identified and quantified with at least two unique peptides and a false discovery rate < 5% in all biological replicates. According to the selected criteria, 38 proteins exhibited significantly different abundance in hSOD1(G93A) astrocytes compared with hSOD1(WT) (Figure 2B). The Student’s *t*-test based statistical significance for such 38 proteins was also confirmed by non-parametric Wilcoxon rank sum test (Appendix A).

The hierarchical cluster analysis of the 38 proteins with altered abundance indicated that hSOD1(WT) and hSOD1(G93A) astrocytes express completely different protein amounts, allowing to divide them into two distinct clusters (Figure 2C).

Gene Ontology (GO) enrichment analysis (performed using the EnrichR webtool) evidenced that processes such as proteolysis and ubiquitination (PSMB4, CTSZ, CTSS, CTSB PSMD8, RAB1A) and phagocytosis (GSN, CORO1A, LRP1, AIF1) are upregulated, while glutathione metabolic processes are downmodulated (GCLC, GSTO1, GLO1, ESD) in hSOD1(G93A) astrocytes (Appendix A).

### 2.3. Validation of TMT-Based Proteomic Results by Western Blot (WB) Analysis

We carried out a validation of TMT-based proteomics results by WB analysis to confirm the modified abundance of selected proteins. Proteins for such a validation study were chosen based on their fold-change or on the relevance of their biological function in relation to ALS pathogenesis.

Following such principles, among the 20 under-expressed proteins, we chose GSTO1, a protein involved in the detoxification of several exogenous stressors [34], also found to be related to fALS [35], and GCLC, the rate-limiting enzyme involved in GSH biosynthesis, consistently with the finding of reduced GSH level in ALS [36,37,38]. Importantly, the GO analysis revealed that both proteins are related to GSH metabolism (Appendix A). The importance of GSH is supported also by the results obtained with the analysis of secreted metabolites (Appendix A, Figure 1C). While WB assessment of protein levels demonstrated a significantly reduced abundance of GCLC (by approximately 20%), only a tendency toward a reduction of GSTO expression was detected in hSOD1(G93A) astrocytes with respect to the hSOD1(WT) counterpart (Figure 3).

Additionally, among the 18 overexpressed proteins, WB analyses confirmed the higher amount of two out of the three lysosomal cysteine proteases were found to be altered in hSOD1(G93A) astrocytes by MS analysis, namely cathepsin B (Cath B, increased by 50%) and cathepsin Z (Cath Z, increased by 120%) (Figure 4).

The WB analyses also confirmed the increased expression (by 20%) of the regulatory subunit 8 of the 26S proteasome (PSMD8) in hSOD1(G93A) astrocytes and the significantly higher amount (increased by 50%) of Gelsolin, a protein playing a central role in the actin dynamics of phagocytes [39], in ALS-related mutant astrocytes (Figure 4).

### 2.4. Computational Identification of Altered TFs in hSOD1(G93A) Astrocytes

We next investigated if the alterations of protein expression in hSOD1(G93A) astrocytes may be related to transcriptional gene regulation guided by specific TFs. A computational analysis was performed to verify if common TFs regulate the expression of genes coding for the altered proteins. To limit false positive results, we used three different algorithms to predict TF binding sites shared among altered genes. Unfortunately, the analysis of the TF binding sites of downregulated genes in hSOD1(G93A) astrocytes did not show common TF binding sites identified by at least two of the used algorithms. Conversely, the analysis of upregulated genes in hSOD1(G93A) astrocytes evidenced a common regulation by at least two algorithms. In particular, Klf4, Zfp423, Mzf1_1-4, Sp1, Insm1, Tcfcp2l1, Creb1, Rel, NF-KappaB, Spi1, Nr1h2::Rxra, Esrrb, Irf2, Nhlh1, Ctcf, Ewsr1-Fli1, Pax5, and Esr2 were identified as regulators of upregulated genes in hSOD1(G93A) astrocytes by two algorithms, while Zfx, Plag1, Ebf1, Nfkb1, and the heterodimer Pparg::Rxra were identified by all three used algorithms (Figure 5 and Appendix A).

### 2.5. Evaluation of TF Alterations in hSOD1(G93A) Primary Astrocytes

To evaluate if the TFs that were predicted to regulate overexpressed genes were altered in hSOD1(G93A) astrocytes, we analysed the expression levels of most TFs predicted to have a binding site in the overexpressed genes in hSOD1(G93A) astrocytes by at least two algorithms by qRT-PCR. Among the predicted TFs in common with the three algorithms used, five of them were screened (early B cell factor 1 (Ebf1), pleomorphic adenoma gene 1 (Plag1), Pparg, Rxra and Zfx). We confirmed the upregulation of four of them, while only Zfx was not found to be significantly upregulated (Figure 6A).

Among those four TFs, the two most upregulated ones were Ebf1 and Plag1, which are involved in the regulation of glial gene expression [40] or in the promotion of neurogenic fate at the expense of astrogliogenic fate [41], respectively.

The qRT-PCR analysis also allowed us to establish that 6 out of the 11 analyzed TFs in common to at least two of the applied algorithms were significantly upregulated and one (Klf4) was downregulated, while four TFs resulted unaltered in the hSOD1(G93A) astrocytes (Figure 6A). Although Creb1 was predicted to be a regulator of overexpressed genes in hSOD1(G93A) astrocytes (Figure 5) and it has been already reported to be related to ALS disease progression [42], no alteration at the transcriptional level was observed. Similarly, WB-based quantification of phosphorylated (active) Creb1 demonstrated that Creb1 activity is not altered in hSOD1(G93A) astrocytes (Figure 6B).

The above reported bioinformatic analysis also indicated that NF-kB may be implicated in protein upregulation in ALS astrocytes. Added to the previous notion that NF-kB is associated with ALS disease progression [43], this finding prompted us to monitor NF-kB activity by checking the phosphorylation state of its regulatory subunits IKKα/β and of IkBα [44] and the translocation of NF-kB p65 subunit from the cytoplasm to the nucleus (using biochemical fractionation assays) in the two astrocyte populations. Such analyses showed that both IKKα/β and IkBα were more phosphorylated (by approximately 100% and 50%, respectively) in hSOD1(G93A) than in hSOD1(WT) astrocytes (Figure 6C), correlating to the increased abundance of NF-kB p65 in the nuclear fraction of hSOD1(G93A) astrocytes (Figure 6D).

## 3. Discussion

It is now recognized that astrocytes are key contributors to disease progression and MN death in ALS [32,45,46]. In the present study, important changes in the cellular proteome and the secreted metabolome were identified in primary spinal astrocytes derived from newborn mice of the hSOD1(G93A) ALS Tg model, which could be useful to identify disease-related mechanisms. Several reports already indicated that the altered abundance of transcripts and proteins in cultured neonatal primary astrocytes was often reproduced in adult/ill ALS mice or ALS patients [26,47,48]. Moreover, the reported selective death of primary MNs when co-cultured with primary hSOD1(G93A) astrocytes supports the statement that primary astrocytes may represent an optimal model to identify molecular events of pathological importance, hampering astrocyte functions and leading to MN death in ALS [49,50].

Proteomics data suggested that disturbances in protein homeostasis and abnormal proteolysis through the ubiquitin–proteasome and/or endosome–lysosome systems occur in ALS astrocytes, where a higher abundance of the non-ATPase regulatory subunit 8 (PSMD8) and of subunit β4 of the 20S catalytic core (PSMB4) of the 26S proteasome was found compared to non-ALS controls. Interestingly, several studies have already linked the up-regulation of the ubiquitin–proteasome system to ALS, both in MN and in glial cells [51,52,53].

Of note, a transversal pathological hallmark of both fALS and sALS is the mislocalization of some proteins and their accumulation in inclusions or aggregates in MNs and surrounding cells, suggesting a disturbance in protein folding and protein quality control. In this context, cells may enhance the expression of proteins involved in degradation processes under proteotoxic stress as adaptive mechanisms.

Moreover, alterations of the endosome–lysosome machinery for intracellular protein degradation recently emerged as a possible pathogenic mechanism in ALS and other neurodegenerative diseases [54,55]. Here we demonstrated that the lysosomal cysteine proteases cathepsin B, S, and Z, whose expression is known to change under inflammatory conditions [56], and other proteins involved in endocytosis and lysosome organization, are overexpressed in ALS astrocytes. Our data support the hypothesis that an upregulation of cathepsin/lysosome functions may play an important role in the deregulation of astrocytes, contributing to the MN degeneration of ALS. Conversely, it may represent an uneven process underpinning astrocyte activation due to the accumulation of protein aggregates.

The notion that unbalanced lysosomal proteolytic activity is not only a hallmark of ALS, but may also represent a first insult contributing to MN death is further supported by previous reports of altered expression or activity of the cathepsin protein family in the glial cells [48,57], skeletal muscle, and MNs of ALS mouse models [58,59,60] and in the MNs and glial cells of ALS patients [61]. In addition to cathepsin proteases, we also found increased levels of the subunit beta of the hexosaminidase enzyme in hSOD1(G93A) primary astrocytes, as previously reported in astrocytes isolated from symptomatic or late-stage disease ALS mice [48]. Hexosaminidase is a lysosomal enzyme involved in the processing of external GM2 gangliosides [62], and the higher amount that we observed might be due to increased phagocytic activity.

This hypothesis is further sustained by our findings that proteins involved in phagocytosis are overexpressed in hSOD1(G93A) astrocytes. Our data imply that the well-recognized ability of astrocytes to phagocyte and eliminate cellular debris [63] is intrinsically increased in the hSOD1(G93A) model.

One of the major achievements of this study is the demonstration of enhanced expression of Ebf1 and Plag1 TFs, which may account (at least in part) for ALS-associated proteome changes.

Ebf1 is a master TF that regulates cell fate specification and is also involved in the expression of glial-specific genes [40]. It plays an essential role as key integrator of signal transduction, inflammation, and the regulation of genes involved in lipid metabolism. Its mutations were previously associated with neurodegenerative disorders [64]. On the other hand, Plag1 is a zinc finger TF playing an important role in the regulation of neurogenic potential [41]. Here, for the first time, to our knowledge, we reported an altered expression of Ebf1 and Plag1 TFs in hSOD1(G93A) astrocytes that may be responsible for the non-cell autonomous mechanisms of MN death in ALS disease.

Regarding the observed NF-kB activation, many of the proinflammatory cytokines, chemokines, enzymes, and adhesion molecules under its transcriptional regulation are already known to be upregulated in ALS [43]. In addition, the NF-kB p65/p50 heterodimer was found to be activated in glia in fALS and sALS patients [65,66] and represents the highest-ranked regulator of inflammation in astrocytes derived from human post-mortem ALS patients [67]. An increased NF-kB activation has been already found during disease progression in the hSOD1(G93A) mouse model, specifically in glial cells [43], although its inhibition did not delay ALS symptoms or death [68].

The transcriptional activity of NF-kB depends on its translocation to the nucleus. This process is allowed by the phosphorylation of IKKb (a subunit of the inhibitor of IkB kinase (IKK) complex) and, consequently, of the inhibitory protein IkBα [44]. In this study we showed an increased phosphorylation of both IKKb and IkBα, and—accordingly—a higher amount of the NF-kB p65 subunit in the nucleus of hSOD1(G93A) astrocytes compared to non-ALS controls, sustaining its enhanced activation in ALS astrocytes.

The proteomics analysis also unveiled that the GSH metabolism was profoundly altered in hSOD1(G93A) astrocytes. Moreover, lower levels of GSH were found in both ALS patients [69,70] and in ALS models [36]. GSH is the major neuronal antioxidant and represents the key component of many detoxification processes, protecting neurons against extended periods of oxidative stress [71]. Indeed, astrocyte-secreted GSH acts as an antioxidant in the extracellular compartment and boosts GSH levels in neurons [71,72]. Accordingly, the continuous consumption of GSH in astrocytes needs to be compensated by a sustained de novo synthesis through the sequential action of two ATP-dependent enzymes, i.e., GCL, and glutathione synthase (GS). While GCL catalyses the rate-limiting step of γ-glutamylcysteine formation from glutamate and cysteine, GS is involved in the ligation of γ-glutamylcysteine to glycine to form GSH [73]. The downregulation of the catalytic subunit of heterodimeric GCL here observed may contribute to the dampened response to oxidative stress, suggested to be a causal agent in ALS. Interestingly, the TF regulating GCL expression, i.e., the nuclear factor erythroid-derived 2-like 2 (Nrf2), was found to be downregulated in post-mortem tissues from ALS patients [74] and in the hSOD1(G93A) mouse model [75], while Nrf2 overexpression in astrocytes had a significant beneficial effect in ALS mouse models [76].

Nonetheless, GCL presents a relatively broad substrate specificity and can bind glutamate with some other amino acids in addition to cysteine, producing the corresponding γ-glutamyl dipeptides [77]. The role of γ-glutamyl dipeptides in cells is not yet fully understood, but they have been detected in both the brain and eyes [78], indicating a potential involvement of these compounds in neuronal homeostasis.

The lower amount of GCLC and/or the lower GSH substrate availability to γ-glutamyltransferase, responsible for the transfer of the γ-glutamyl moiety to any amino acid [78], may thus account for the lower amount of the γ-glutamyl-tyrosine dipeptide detected in the CM of hSOD1(G93A) astrocytes by the metabolomic analysis.

The minor amount of secreted cysteine-S-sulfate may be also attributed to altered GSH metabolism. In mammals, GSH may be obtained also by the action of GSH reductase (GR), which recycles S-glutathionyl-L-cysteine and sulphate to originate cysteine-S-sulfate and GSH. GR was not detected by the proteomic analysis reported here, but previous studies reported that lower amounts of this enzyme were responsible for increased oxidative stress in erythrocytes of sALS patients [79].

GSH is also used in astrocytes for the detoxification of xenobiotics and endogenous compounds by GSH-S-transferases (GST), which generate GSH conjugates that are efficiently exported from the cells by multidrug resistance proteins [71]. Here, we reported a lower amount of the enzyme GSTO-1, suggesting a major vulnerability of ALS astrocytes to the toxic effects of electrophilic compounds and organic peroxides. In addition, GST downregulation was already reported in ALS patients [80], while GSTO-2 overexpression in a *Drosophila* model of ALS recovered defective phenotypes, supporting the hypothesis that it represents a potential therapeutic target for ALS [81]. In summary, both our data and previous reports support the notion that ALS astrocytes have a lower capacity for GSH-dependent detoxification processes, thereby hampering the astrocytes protective effects on neighbouring neurons against the action of oxidants and toxins.

Moreover, metabolomics analysis allowed us to identify some FFA overrepresented in the secreted metabolome of hSOD1(G93A) astrocytes compared to the hSOD1(WT) counterpart. At the same time, we found halved amounts of two intermediates of glycerolipid metabolism (i.e., galactosyl-glycerol and phosphatidic acid). These results highlighted that alterations in lipid metabolism/signaling processes occur in ALS astrocytes, which are consistent with previously reported data [24,82,83,84]. Interestingly, lipid handling from astrocytes to neurons plays an essential role not only as a source of energy, but also for mediating neural circuit formation and function [85]. This process is also carried out via the action of released phosphatidic acid by astrocytes that is important for the activation of lipid-gated ion channels and have long been linked to neurite outgrowth [86]. The minor amount of phosphatidic acid released by hSOD1(G93A) astrocytes may underscore an impairment in the astrocyte-neuron crosstalk, possibly contributing to MN death in ALS.

Finally, here we observed alterations in the amount of molecules participating in the biosynthesis/metabolism of few amino-acids in ALS astrocytes, as previously reported [22,23,24]. For example, we found altered intermediates of the biosynthesis of glutamate, a well-known excitatory neurotransmitter but also a key contributor for GSH synthesis, and whose disturbance was reported not only in ALS mice but also in ALS patients [23,87].

## 4. Materials and Methods

### 4.1. Animals

In this study, we used the hSOD1(G93A) (B6SJL-Tg(SOD1*G93A)1Gur/j) and the hSOD1(WT) (B6SJL-Tg(SOD1)2Gur/J) mice overexpressing the human ALS-related mutant G93A or the WT isoform of SOD1 protein, respectively [13,88]. Both strains were purchased by Jackson Laboratories. To maintain the colonies, hemizygous Tg males were bred with non-Tg B6SJLF1/J hybrid females. Offspring positive for the hSOD1(G93A) or hSOD1(WT) transgenes were identified through genotyping PCR analysis, as previously described [89]. All aspects of animal care and experimentation were carried out in compliance with European and Italian (D.L. 26/2014) laws concerning the care and use of laboratory animals. The authors’ Institution has been accredited for the use of experimental mice by the Italian Ministry of Health (Authorization N. 305/2017-PR, released on 6 April 2017) and by the Ethical Committee of the University of Padova.

### 4.2. Primary Cultures of Murine Spinal Cord Astrocytes

Primary cultures of spinal astrocytes were prepared and cultured as previously reported [80]. Briefly, cultures were obtained from spinal cords explanted from hSOD1(WT) and hSOD1(G93A) newborn mice (1–2 days old). After dissociation of tissues, cells were suspended in a culture medium (minimal essential medium, L-glutamine 4 mM, glucose 0.3% (*w*/*v*), penicillin 100 U/mL, streptomycin 100 μg/mL) supplemented with fetal bovine serum (FBS) 20% (*v*/*v*) and cultured for two weeks in 25 cm^2^ flasks. Upon reaching the confluence, primary astrocytes were re-plated at the optimal density of 40,000 cells/cm^2^ in 12-well plates. Cells were maintained for 96 h in culture medium supplemented with FBS 10% (*v*/*v*). At this stage, culture purity was estimated to be at least 95%, by calculating the percentage of glial fibrillary acidic protein-positive cells in the total population of cultured cells (Hoechst stained), as already reported [89]. For metabolomics and proteomics analyses, astrocytes extracted from three pups (for each hSOD1 genotype) were cultured separately in three different T25 flasks. Then, cells deriving from each flask were re-seeded in 5 different 4 cm^2^ wells, giving rise to 15 different replicates. After 96 h, the CM of each well was separately collected for the large-scale secreted metabolome analysis, giving rise to a total of 15 biological/technical replicates. Conversely, for proteomics analysis, cells from 5 wells originating from a single animal were pooled together again and lysed, thus generating three different biological replicates.

### 4.3. Untargeted Metabolomic Analysis of the Secretome of Primary Cultures of Astrocytes

A metabolomics analysis of 15 CM derived from 3 primary cultures of hSOD1(G93A) or hSOD1(WT) spinal astrocytes was performed as in Gomez et al. [90]. After 96 h of culturing, CM were collected and centrifuged (10 min, 16,000× *g*, 4 °C) to remove floating dead cells and debris. To precipitate the protein fraction, CM (100 µL) were diluted with 400 µL of ice-cold methanol and stored for 16 h at −80 °C. Samples were spun (15 min, 16,000× *g*, 4 °C) and the supernatants (150 µL) were collected and dried using a speed vacuuming system. Dried samples were reconstituted with 75 μL of 50/50 acetonitrile/water solution (*v*/*v*) and centrifuged (5 min, 16,000× *g*, 4 °C). Then, the samples were subjected to metabolomics profiling by HILIC analysis. A QC sample was prepared by mixing an equal volume of supernatant from each sample under investigation and by adding a mixture of labelled internal standards (leucine-5,5,5-D3, L-tryptophan-2,3,3-D) at a final concentration of 1 ng/μL.

LC-HRMS was performed on an Ultimate 3000 UHPLC system (Ultimate 3000, Thermo Scientific, Waltham, MA, USA) interfaced to a quadrupole-Orbitrap mass spectrometer (Q-Exactive, Thermo Scientific, Waltham, MA, USA) equipped with a heated electrospray ionization source. Each sample (5 μL) was injected onto a Zic-HILIC column (100 × 2.1 mm; 3.5 µm particle size, Merck, Darmstadt, Germany) kept at 30 °C and equipped with a guard column. The chromatographic separation of metabolites was achieved in 25 min, at a flow rate of 0.3 mL/min, using 2.5 mM acetic acid, 2.5 mM ammonium acetate in water, and pH 6.0 (A) and acetonitrile (B) as elution solvents. The composition of the mobile phase (expressed as A:B ratio) during the chromatographic separation was as follows: 5:95 from 0 to 2 min, increased linearly to reach 60:40 at 15 min, kept unchanged until 19 min, and finally linearly decreased to reach 5:95 (initial conditions) at 19.5 min, and kept until the end of the run (25 min) to re-equilibrate the system. MS data were acquired in full scan mode at resolving power of 70,000 full width at half maximum at 200 *m*/*z*. Source parameters were as follows: spray voltage, 2.5 kV and −2.5 kV in positive or negative polarity, respectively; capillary temperature, 325 °C; sheath gas flow rate, 40 arbitrary units (a.u.); auxiliary gas flow rate, 10 a.u.; S-lens voltage, 50 V; in-source fragmentation voltage, 0; heater temperature, 325 °C. UHPLC-HRMS performance was evaluated by injecting QC samples throughout the analytical run (as the first and last samples, and every six samples, for a total of six QC injections). Astrocytes CM metabolomics profiling was carried out randomizing the injection order of CM samples.

Obtained UHPLC-HRMS data were processed using Compound Discoverer software (Thermo Fisher Scientific, Waltham, MA, USA, version 2.1) to perform alignment of chromatographic peaks, integration of compound peaks and normalization of area values, as previously described [91]. The acceptability of the acquired data was evaluated by checking the consistency of the chromatographic retention time and the mass accuracy of chromatographic peaks (<10 ppm) of labelled internal standards in the six within-run QC samples.

### 4.4. Metabolomics Data Analysis

Two data matrices resulting from the metabolomics profiling carried out either in negative or in positive ionization polarity were analysed. To visualize sample distribution and highlight similarities between sample groups, unsupervised multivariate PCA was performed on mean centered data using the SIMCA-P software (version 13.0, Umetrics, Sweden).

The determination of altered metabolites was performed as previously reported [91]. Briefly, chromatographic peaks derived from the two datasets were manually inspected and filtered considering acceptable for further analyses metabolites whose relative standard deviation in QC samples was <30%.

Normalized area values derived from chromatographic peaks of the detected metabolites were used to compare the relative amounts between hSOD1(G93A) and hSOD1(WT) and expressed as ratios. The statistical significance was assessed using the Wilcoxon non-parametric test comparing hSOD1(G93A) and hSOD1(WT) CM. To visualize the magnitude and statistical significance of altered metabolites, processed data were represented in a Volcano plot by plotting for each compound log_2_ (ratio) on the *x*-axis, and -log_10_ (*p*-value) on the *y*-axis. Significantly changed metabolites were filtered by applying a *p*-value < 0.05 and a ratio of 1.5 as cut-off parameters.

### 4.5. Astrocytes Lysis and Protein Extraction

For proteomic and WB analyses, astrocytes were washed twice with phosphate buffer saline (PBS), lysed in 60 µL of buffer O (glycerol 10% (*w*/*v*), sodium dodecyl-sulphate (SDS) 2% (*w*/*v*), Tris/HCl 62.5 mM, pH 6.8, and cocktails of protease and phosphatase inhibitors (Roche, Basel, Switzerland)) and centrifuged (10 min, 14,000× *g*, 4 °C) to remove cell debris. Protein concentration was determined in the supernatants with a Lowry assay kit (Sigma-Aldrich, St. Louis, MO, USA), according to the manufacturer’s instructions. For WB analyses, samples were diluted to an equal protein concentration using reducing buffer O (i.e., added with dithiothreitol (DTT) (50 mM, final concentration (f.c.) and blue di-bromophenol (0.1% (*w*/*v*), f.c.)), and boiled (5 min).

### 4.6. Proteomics Profiling of Primary Astrocytes

Lysates from three cultures of hSOD1(WT) and hSOD1(G93A) primary astrocytes were processed according to the filter-aided sample preparation protocol [92]. Briefly, protein extracts (50 μg in 50 μL) were mixed with 200 μL of 8 M urea in 100 mM triethylammonium bicarbonate (TEAB) (Sigma-Aldrich, St. Louis, MO, USA). Samples were then loaded in filters with a molecular weight cut-off of 10 kDa (Sartorious, Goettingen, Germany) and centrifuged (15 min, 14,000× *g*, 25 °C). Disulphide bonds were reduced and alkylated using 200 μL of 25 mM DTT (45 min, 55 °C) and, after centrifugation, 200 μL of 55 mM iodoacetamide (45 min in the dark, RT) (both diluted in 100 mM TEAB), respectively. After removal of the solution by centrifugation, protein digestion was carried out by incubating filters (18 h, 37 °C) with a sequencing grade modified trypsin (Promega, Madison, WI, USA) solution (90 μL in 100 mM TEAB, pH 8.0) to a final enzyme:protein ratio of 1:25 (*w*/*w*). The released peptides were collected by centrifugation (10 min, 14,000× *g*). To improve recovery, filters were washed with 50 μL of 100 mM TEAB and centrifuged again.

To perform quantitative MS, peptides were labelled with 6-plex TMT reagents (Thermo Scientific, Waltham, MA, USA), according to the manufacturer’s instructions. To this purpose, peptide samples from three hSOD1(WT) and three hSOD1(G93A) biological replicates were labelled with six TMT reporter ions (with *m*/*z* of 126.1, 127.1, 128.1 and 129.1, 130.1, 131.1 Thomson (Th), respectively). Samples were then equally mixed, desalted using C18 BioPure spin columns (Nest Group, Ipswich, MA, USA), and eluted into five fractions by strong cation exchange fractionation (SCX) by increasing the ammonium formate concentration (i.e., 100 mM, 200 mM, 300 mM, 400 mM, 500 mM), as previously described [93]. SCX fractions were then purified by means of C18 BioPure spin columns using a standard protocol and dried using a stream of nitrogen at room temperature. To proceed with MS analysis, the dried fractions were resuspended in 100 μL of 0.1% formic acid.

### 4.7. UHPLC-HRMS/MS Analysis

MS analysis was performed using a hybrid quadrupole-Orbitrap Q-Exactive (Thermo Fisher Scientific, Waltham, MA, USA) mass spectrometer, coupled to an Ultimate 3000 UHPLC system (Thermo Fisher Scientific, Waltham, MA, USA). Aliquots containing 5 µg of digested proteins (10 µL) were separated on a reversed phase analytical column (Aeris peptide C_18_, 150 × 2.1 mm, 2.6 μm, Phenomenex, Torrance, CA, USA) kept at 30 °C. The mobile phases were water (A), and acetonitrile (B), both containing 0.1% formic acid (*v*/*v*). Peptide separation was carried out at a flow rate of 200 μL/min using the following gradient: from 0 to 1 min the composition of mobile phase (A:B) was kept constant at 97.5:2.5. Then the composition was changed linearly to reach 30:70 at 20 min, and changed again to reach 50:50 at 24 min. Finally, the composition was increased linearly to 5:95 at 26 min and kept unchanged until 30 min to wash the column. The system was re-equilibrated to the initial condition (97.5:2.5) in 0.5 min and maintained for 4.5 min until the end of the chromatographic run (35 min).

Instrumental conditions were as previously described [94]. Each SCX fraction was analysed twice to increase the number of identified peptides, by applying a static exclusion list (containing peptides identified in the first run) that was applied to the second UHPLC-HRMS/MS analysis.

### 4.8. Untargeted MS Data Analysis

MS raw files were analysed with a MudPIT protocol using the Proteome Discoverer 1.4 software (Thermo Fisher Scientific, Waltham, MA, USA) interfaced to a SEQUEST HT search engine (Thermo Fisher Scientific, Waltham, MA, USA). Spectra were searched against the UniProt *Mus musculus* database. Trypsin was selected as the enzyme with up to 2 missed cleavages allowed. Peptide and fragment mass tolerances were 10 ppm and 0.1 Da, respectively. TMT 6-plex (N-term and Lys) and carbamidomethyl-cysteine were set as fixed modifications, while methionine oxidation was selected as variable modification. Only unique peptides were considered for quantification purposes, which was performed directly by the software, setting the tolerance of the reporter mass to 20 ppm. By using the algorithm Percolator proteins were considered correctly identified if at least 2 unique peptides were quantified with high confidence (q-value < 0.05). Principle of maximum parsimony was applied to grouped proteins. The quantification was performed by the software normalizing the intensity of each reporter ion to that of the hSOD1(WT) sample (126.1 Th). Sample distribution was visualized by performing PCA analysis using the SIMCA-P software (version 13.0, Umetrics, Sweden), while the determination of altered proteins was performed as previously reported [94]. Indeed, data were exported in an Excel spreadsheet and used for further analysis. The fold-change of a given protein was estimated as the mean ratio of hSOD1(G93A)-to-hSOD1(WT) values and subjected to a two-tailed T-test (*p* ≤ 0.05). A ratio of hSOD1(G93A)-to-hSOD1(WT) >1.33 or <0.77, was set as the threshold for increased and decreased abundance, respectively. The graphical visualization of altered proteins was allowed by plotting each protein in a Volcano plot and using the log_2_ of ratio values (for the *x* axis), and the -log_10_ of *p*-value (for the *y*-axis). In order to confirm the statistical significance by a non-parametric test, the Wilcoxon rank sum test was also applied to all proteins resulting altered using the Student’s *t*-test. The obtained *p*-values were reported in Appendix A.

Clusterization of astrocytes samples was performed by the construction of the heat map with hierarchical clustering based on the Pearson correlation with average linkage using normalized area values of the 38 altered proteins (MeV software package (version 4.9.0, SourgeForge, San Diego, CA, USA)). GO enrichment analysis on differentially expressed proteins between hSOD1(G93A) and hSOD1(WT) was performed with EnrichR webtool [95,96,97] using default parameters.

### 4.9. Subcellular Fractionation

For fractionation experiments, astrocytes (approximately 600,000 cells) from three different primary cultures for each genotype were rinsed once with ice-cold PBS, lysed in 100 µL of chilled Buffer A (containing 15 mm Tris-HCl pH 7.9, 10 mM KCl, 0.1 mM EDTA, 0.1 mM EGTA, 1 mM DTT and a protease inhibitor cocktail (Roche, Basel, Switzerland)) and incubated on ice for 5 min. Then, an equal volume of Buffer A with 0.6% NP-40 (*v*/*v*) was added to the solution, and the samples were incubated for an additional 5 min on ice and stirred occasionally. The suspension was centrifuged at 1000× *g* for 10 min at 4 °C. The supernatant was set aside as the cytosolic fraction. The pellet was gently resuspended in Buffer A and re-centrifuged to remove any cytosolic contamination. Afterward, to purify the nuclear fraction, the pellet was resuspended in Buffer B (containing 15 mm Tris-HCl pH 7.9, 10 mM KCl, 1.5 mM MgCl_2_, 0.1 EGTA, 0.4 M NaCl, 1 mM DTT, 5% *w/v* glycerol and a protease inhibitor cocktail (Roche, Basel, Switzerland) and incubated for 30 min on ice (vortexed every 5 min during the incubation) and then centrifuged (20,000× *g*, 10 min, 4 °C). Quantification of protein content in the cytosolic and nuclear fractions was performed using the Lowry assay kit (Sigma-Aldrich, St. Louis, MO, USA). The equivalent of 10 µg of total proteins from each sample was then analysed through SDS-PAGE and WB analysis.

### 4.10. WB Analysis and Antibodies

Protein samples (10 µg) were separated by SDS-PAGE (Mini-Protean TGX precast gels, Bio-Rad Laboratories) and then electrophoretically transferred onto polyvinylidene difluoride (PVDF) (Millipore, Burlington, MA, USA) or nitrocellulose membranes (Bio-Rad Laboratories, Hercules, CA, USA). After incubation with a blocking solution (BS) (3% (*w*/*v*) BSA (Sigma-Aldrich) in Tris Buffered solution (TBS) added with 0.1% (*w*/*v*) Tween-20 (TBS-T)) (1 h, 25 °C), membranes were probed (overnight, 4 °C) with the required primary antibodies (ab) diluted in BS. After three washings with TBS-T, membranes were incubated (1 h, 25 °C) with horseradish peroxidase-conjugated anti-mouse or anti-rabbit IgG, depending on the primary antibody host specification (Sigma Aldrich, cat. n. A9044 and A0545, respectively). Immunoreactive bands were visualized and digitalized by means of an UVItec imaging system (Eppendorf, Hamburg, Germany), using an enhanced chemiluminescence reagent kit (EMD Millipore, Burlington, MA, USA). To ensure equal protein loading, nitrocellulose or PVDF membranes were stained with Ponceau or Coomassie brilliant blue solution, respectively. To determine the relative protein abundance, densitometric analyses were performed by normalizing the intensity of each immunoreactive band to the optical density of the corresponding Coomassie blue or Ponceau-stained lane [98].

The following primary monoclonal antibodies were used (dilutions are in parentheses): mouse anti-Cath B (1:100; Santa Cruz Biotechnology, Heidelberg, Germany, cat. n. sc-365558); mouse anti-PSMD8 (1:100; Santa Cruz Biotechnology, Heidelberg, Germany, cat. n. sc-514053); rabbit anti-GCLC (1:1000; Elabscience, Wuhan, China cat. n. E-AB-52359); rabbit anti-Cath Z (1:1000; Elabscience, Wuhan, China, cat n. E-AB-13977); rabbit anti-Gelsolin (1:1000, Genetex, Irvine, CA, USA, cat. n. GTX114078); mouse anti-phosphorylated (on Ser133) Creb1 (1:1000, Cell Signaling Technology, Danvers, MA, USA, cat. n. 9196s); rabbit anti-Creb1 (1:1000, Cell Signaling Technology, Danvers, MA, USA cat. n. 9197s), rabbit anti GSTO1 (1:1000, Genetex, Irvine, CA, USA, cat. n. GTX105655); rabbit anti-phosphorylated (on Ser176) IKKα (1:1000, Cell Signaling Technology, Danvers, MA, USA cat. n. 2078); rabbit anti-phosphorylated (on Ser32) IkBα (1:1000, Cell Signaling Technology, Danvers, MA, USA, catn. 2859); rabbit anti- IkBα (1:1000, Cell Signaling Technology, Danvers, MA, USA, cat. n. 4814); rabbit anti-IKKα (1:1000, Cell Signaling Technology, Danvers, MA, USA, cat. n. 2682); rabbit anti-NF-kB p65 (1:1000, Cell Signaling Technology, Danvers, MA, USA, cat. n. 8242); rabbit anti Histone3 (1:1000, Abcam, Cambridge, UK, cat. n. ab18521); mouse anti-tubulin (1:16,000, Sigma Aldrich, St. Louis, MO, USA, cat. n. T8203.)

### 4.11. Promoters Analysis

The oPOSSUM web-based system (v 3.0) [99], pScan [100], and TFM explorer [101] were used for identifying over-represented, conserved binding sites for transcription factors (TF) in the promoter region of the target genes. All algorithms were used with the JASPAR vertebrate matrices [102]. Following parameters used for each algorithm. oPOSSUM: minimum specificity, 8 bits; conservation cut-off, 0.40; matrix score ≥ 80%; 5000 bp upstream and 5000 bp downstream the transcription start site; pScan: select region −959 +50; TFM explorer: 1500 bp upstream and 200 bp downstream the transcription start site; Number of clusters to display 25; maximum *p*-value 0.001; ratio (density of clusters) 3. To select TF from oPOSSUM and pScan results, we used the z-score higher than 1.96, instead TFM explorer showed only the top 25 significant clusters. This is one limit of this web software that does not allow to see more than 25 clusters. Venny 2.1 web tool [103] was used to design Venn diagrams.

### 4.12. Total RNA Extraction and Retrotranscription

Total RNA was isolated from primary spinal astrocytes (160,000 cells) using the Trizol Reagent (Thermo Fisher Scientific), according to the manufacturer’s protocol. Briefly, after cell lysis with Trizol (500 μL), chloroform (100 μL) was added to the homogenate to separate the RNA-containing aqueous phase. Thereafter, RNA was precipitated by adding an equal volume of isopropanol to the aqueous phase. Qualitative and quantitative RNA measurements were assessed both by spectrophotometer and denaturating gel electrophoresis analyses. cDNA was synthetized starting from 1 µg of total RNA using the High Capacity cDNA Reverse Transcription kit (Thermo Fisher Scientific, Waltham, MA, USA), according to the manufacturer’s protocol.

### 4.13. Primer Design and Quantitative Real-Time PCR (qRT-PCR) Analysis

For each target gene monitored in qRT-PCR experiments (whose sequences were downloaded from the NCBI Gene database), primer sequences were designed using the Primer 3 Plus software by setting default parameters (https://www.bioinformatics.nl/cgi-bin/primer3plus/primer3plus.cgi accessed on 22 June 2021). To exclude the formation of primer dimers and hairpins, primer sequences were bioinformatically screened with the IDT Oligo Analyzer tool (Integrated DNA Technologies, Inc., Coralville, Iowa, USA). Moreover, primer pairs specificity was evaluated by the *in-silico* PCR tool implemented in the UCSC Genome Browser. Primer pairs chosen are listed in Appendix A. The amplification by qRT-PCR was accomplished in a final volume (f.v.) of 5 μL per well, including 1 µL of EvaGreen qPCR Mix Plus (Solis BioDyne), 0.1 μL of forward primer (10 μM), 0.1 μL of reverse primer (10 μM), 8 ng of the qPCR cDNA template, and water to f.v. The reactions were performed in the CFX thermocycler (BioRad) with the following program: 10-min-denaturation at 95 °C, 40 cycles amplification (95 °C for 15 s; 60 °C for 20 s; 72 °C for 45 s, annealing and extension); 3 min at 72 °C and dissociation curve.

Six different genes (Actin; TATA binding protein (Tbp); RNA Polymerase II Subunit A, (Polr2a); Glyceraldehyde 3-phosphate dehydrogenase, (Gapdh); ribosomal protein lateral stalk subunit P0, (Rplp0); and ribosomal protein L32, (Rpl32)) were tested and categorized according to their coefficient of variation to choose the best reference genes for accurate normalisation of data (Appendix A). Target-gene mRNA levels were quantified according to the ΔCt method and using the two best reference genes (Polr2a and Tbp; see Appendix A) as housekeeping genes. Data were represented as expression value relative to the average expression of the gene among samples.

### 4.14. Statistical Analyses

Statistical analyses were performed using Prism 7 (GraphPad Software) and Microsoft Excel (Microsoft Corporation) software. The number of biological replicates (n) reported in figure legends indicates the number of different primary cultures of hSOD1(WT) and hSOD1(G93A) astrocytes used in each experiment. Statistics was based on unpaired two-tailed Student’s *t*-test, non-parametric Mann–Whitney U test or Wilcoxon rank sum test, depending on the experiment, as indicated in the figure legends, with a *p*-value < 0.05 being always considered statistically significant.

## 5. Conclusions

Here we reported a global comparative analysis of the proteome and metabolite secretome of primary astrocytes derived from hSOD1(G93A) (a well-recognized ALS animal model) and hSOD1(WT) Tg mice (used as the control). Our results showed the alterations of proteome and secretome in ALS astrocytes. However, since most of the altered proteins are not metabolic enzymes, it is fair to recognize a limited correspondence between proteome alterations and secreted metabolome. Nonetheless, our findings support a possible correlation between altered proteome and secreted metabolome in GSH metabolism and signalling in hSOD1(G93A) astrocytes. Furthermore, the metabolism of FA and processes involving degradative processes mediated by the proteasome or by lysosomal pathways appear to be compromised in astrocytes derived from ALS mice. This metabolic and proteostatic imbalance may be responsible for altered astrocyte functions, prejudicing the interplay with neighboring neurons and contributing to neuronal death in ALS. To conclude, as already stated, our study on the here proposed ALS model highlighted metabolic and proteomic dysregulations that may underlie astrocyte-related mechanisms in ALS pathogenesis. Further studies in adult/ill ALS mice and in human ALS patients will be necessary to further validate such findings.

## Figures and Tables

**Figure 1 ijms-22-07028-f001:**
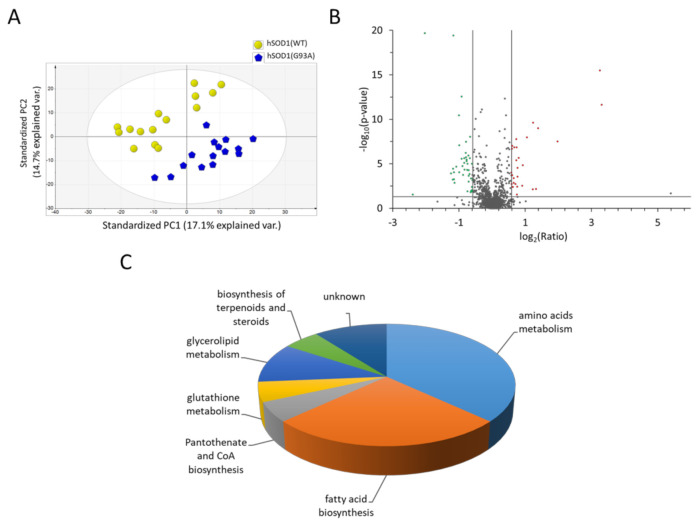
UHPLC-HRMS approach unveils the alteration of metabolite abundance in the CM of hSOD1(G93A) primary spinal astrocytes. (**A**) Principal component analysis (PCA) score plot of CM metabolomics from hSOD1(WT) (yellow circles) and hSOD1(G93A) (blue pentagons) primary astrocyte cultures. Data were acquired by HILIC-HRMS in negative ionization mode. The percentage of variance explained by the first two principal components (PC1 and PC2) is reported in the PCA score plot. Ellipse: Hotelling’s T2 (95%). (**B**) Volcano plot depicting the distribution of the 779 compounds detected in the negative ionization polarity. The abundance of metabolites (expressed as the log_2_ (G93A vs. WT ratio)) and the statistical significance (expressed as the −log_10_ (*p*-value)) were used as representing criteria for the *x*- and *y*-axis, respectively. A ratio <0.66 and >1.5, and *p*-values < 0.05 were used as cut-off parameters (grey circles, non-altered metabolites; green circles, underrepresented metabolites in hSOD1(G93A); red circles, overrepresented metabolites in hSOD1(G93A)). (**C**) Pie chart representing the distribution of significantly altered KEGG pathways between hSOD1(WT) and hSOD1(G93A) astrocytes (other details are reported in Appendix A).

**Figure 2 ijms-22-07028-f002:**
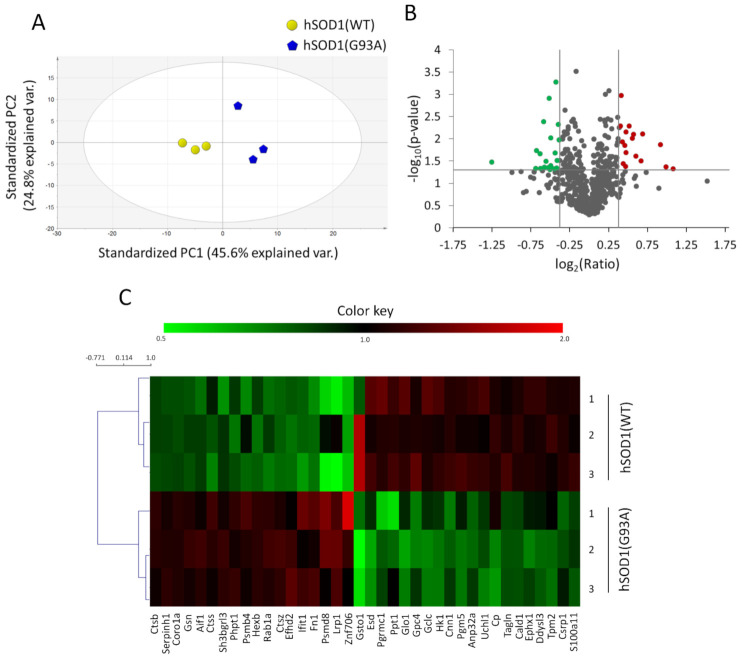
TMT-MS approach unveils protein abundance alterations in hSOD1(G93A) primary spinal astrocytes. (**A**) PCA score plot of the proteomic data of hSOD1(WT) (yellow circles) and hSOD1(G93A) (blue pentagons) primary astrocytes. The percentage of variance explained by the first two principal components (PC1 and PC2) is reported in each axis of the plot. Ellipse: Hotelling’s T2(95%). (**B**) Volcano plot distribution of the 640 quantified proteins, mapping the most significant 20 downregulated (green circles), and 18 upregulated (red circles) proteins in primary spinal cord astrocytes derived from hSOD1(G93A) mice compared to hSOD1(WT) matched controls, satisfying the criteria of log_2_ (fold-change) value < 0.77 e > 1.3 and *p*-values < 0.05. (**C**) Cluster heat map diagram of the 38 proteins whose expression levels were significantly altered in primary spinal cord astrocytes from hSOD1(G93A) mice compared to hSOD1(WT) matched samples. Hierarchical clustering tree indicates primary cultures from hSOD1(G93A) spinal cord differ from hSOD1(WT). Relative protein expression levels are reported by a pseudocolor scale, with red denoting higher and green denoting lower protein expression in hSOD1(G93A) astrocytes compared to the hSOD1(WT) counterpart, respectively.

**Figure 3 ijms-22-07028-f003:**
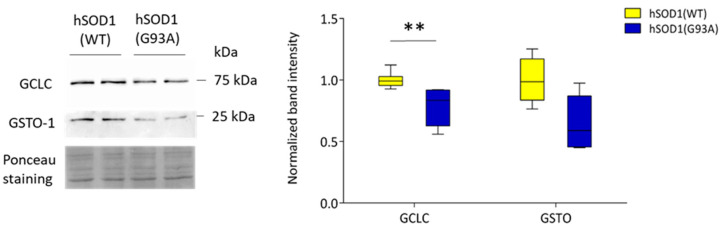
GCLC and GSTO1 are under-expressed in hSOD1(G93A) primary astrocytes-WB validation of GCLC and GSTO1 abundance in hSOD1(G93A) and hSOD1(WT) primary cultures of spinal cord astrocytes. The left panel shows a representative WB. Ponceau staining was used to validate the loading control. The right panel reports the densitometric analysis of GCLC and GSTO-1 immunoreactive bands in the different astrocyte samples normalized to the optical density of the corresponding Ponceau-stained lane. Here and after, black lines in the box plot indicate the median value, while whiskers above and below the box indicate the 5 and 95 percentiles, respectively; *n* = 5 (biological replicates), ** *p* < 0.01 (Mann–Whitney U test).

**Figure 4 ijms-22-07028-f004:**
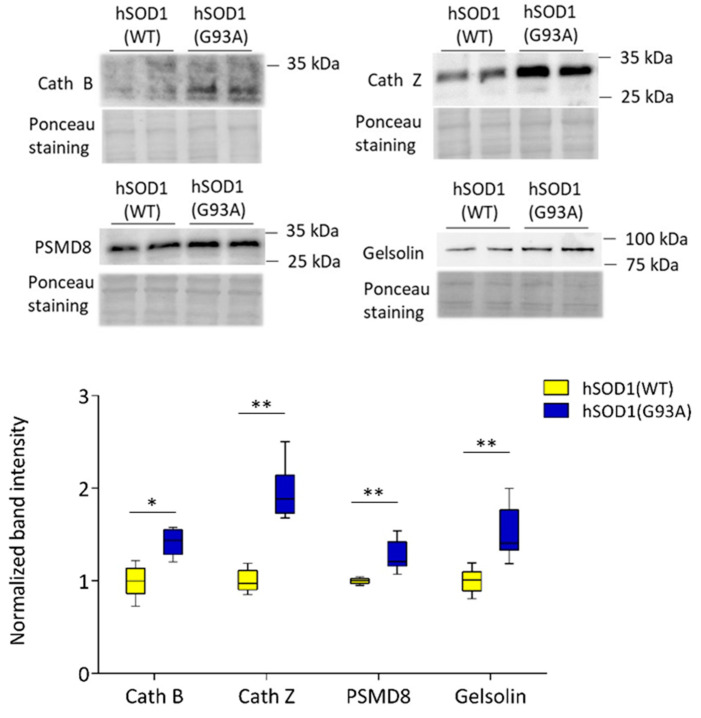
WB validates the hSOD1(G93A)-dependent overexpression of proteins involved in proteolysis. WB validation of Cath B, Cath Z, PSMD8, and Gelsolin abundance in hSOD1(G93A) and hSOD1(WT) primary cultures of spinal cord astrocytes. In the left panel, representative WBs and the corresponding Ponceau stained membranes are reported. In the right panel, the densitometric analysis of CathB, CathZ, PSMD8, and Gelsolin immunoreactive bands in the different astrocyte samples normalized to the optical density of the corresponding Ponceau-stained lanes are reported. *n* = 5, * *p* < 0.05, ** *p* < 0.01 (Mann–Whitney U test). Other details are as in the legend of Figure 3.

**Figure 5 ijms-22-07028-f005:**
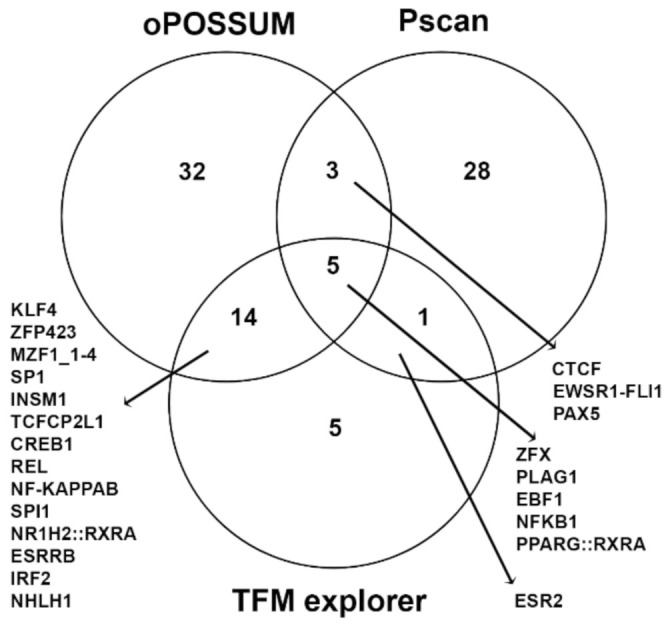
TFs predicted by computational analysis. The Venn diagram reports the number and name of TFs identified by oPOSSUM, Pscan, and TFM explorer algorithms, and whether they are unshared or in common among two or three algorithms.

**Figure 6 ijms-22-07028-f006:**
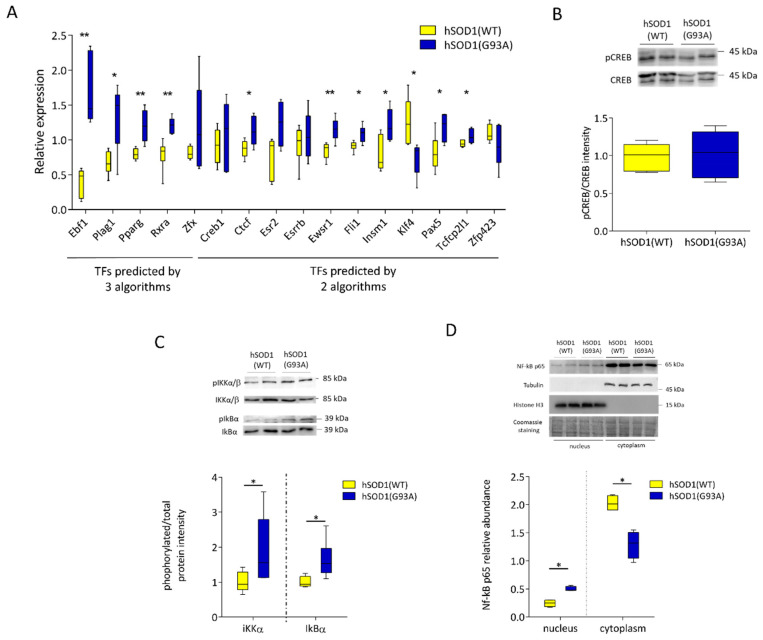
hSOD1(G93A) primary astrocytes showed alterations in the expression or activity of some TFs. (**A**) qRT-PCR uncovered a strong alteration of Ebf1 and Plag1 abundance in hSOD1(G93A) astrocytes. TFs mRNA relative abundance was obtained by qRT-PCR analysis, using primer pairs specific for each target gene (Appendix A). The mRNA amount of hSOD1(WT) (yellow bar) or hSOD1(G93A) (blue bar) was normalized to the mRNA amount of the Polr2a reference gene in the corresponding sample. Box plots represent the distribution of expression values relative to the average expression of the gene among samples. Similar results were obtained using TBP as reference gene. Data are mean ± SD; *n* = 3; * *p* < 0.05; ** *p* < 0.01 (Mann–Whitney U test). (**B**) Creb1 is not activated in hSOD1(G93A) astrocytes. Protein lysates derived from hSOD1(WT) and hSOD1(G93A) primary spinal astrocytes were analysed by WB using antibodies against phospho-Creb1 or total Creb. The upper panel shows a representative WB, while the bar diagram shows the ratio between pCreb and total Creb immunoreactive band intensities. *n* = 5. (**C**,**D**) NF-kB is more activated in hSOD1(G93A) astrocytes. (**C**)—Proteins from hSOD1(WT) and hSOD1(G93A) primary spinal astrocytes were analysed by WB for the phosphorylation state of IKKα/β and IKBα. Bar diagrams show the ratio between pIKKα/β and IKKα/β or pIKBα and IKBα band intensity, respectively. *n*= 4, * *p* < 0.05, (Mann–Whitney U test). (**D**) WB analysis of NF-kB p65 subunit abundance in the nuclear and cytoplasmic fraction of primary astrocytes derived from hSOD1(WT) and hSOD1(G93A) mice. Tubulin and histone H3 proteins were used as internal controls for the cytoplasmic and nuclear fractions, respectively. Coomassie blue staining was used to validate the loading control. The bar diagram reports the densitometric analysis of the nuclear and cytosolic p65 subunit band intensity normalized to the corresponding Coomassie-stained lane. *n* = 3; * *p* < 0.05 (Mann–Whitney U test). Other details are as in Figure 3.

## Data Availability

MS proteomics data have been deposited via the PRIDE partner repository with the dataset identifier PXD026576.

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
