# Peer review of "Perturbations of the Proteome and of Secreted Metabolites in Primary Astrocytes from the hSOD1(G93A) ALS Mouse Model"

_ijms, 2021, doi:10.3390/ijms22137028_

Round 1

Reviewer 1 Report

In the manuscript, the authors reported alterations of 682 proteome and secretome in ALS astrocytes in the primary astrocytes derived from hSOD1(G93A) compared with hSOD1(WT) Tg control mice. Overall, the experiments are well performed and the results are potentially interesting. Therefore, I have only minor comments.

  • Primary cultures of spinal astrocytes were prepared from spinal cords explanted from newborn mice (1-2 days old). Do the authors think that the gene alterations observed in the study could also occur in the matured ALS mice after the onset? This point should be clarified.

Reviewer 2 Report

This is an interesting metabolomics and proteomics analysis performed on primary astrocytes cultures generated from SOD1 mutant (ALS) vs control mice.

The study merit attention and could be of interest, notably as a source of original data. There are however major issues that need to be adressed, in particular regarding the statistical analyses. I personally prefer a study without statistical significance, only trends, rather than a paper with wrong analyses. Being not able to reach significance is not necessarily a problem as far as it is clearly stated and that conclusions are coordingly a bit dosn played.

Here are my detailed comments:

When the authors state that « Metabolomics analysis of 15 CM derived from 3 primary cultures of hSOD1(G93A) 439 or hSOD1(WT) spinal astrocytes was performed as in Gomez et al. » does this mean tha only a total of 3 primary cultures were performed or is this 3 per condition (hSOD1(G93A) 439 or hSOD1(WT)) ? This should be more clearly stated along with the existence (or not) of technical replicates.

Idem for proteomics analyses. In addition, student’s t test cannot be used to compare groups of n = 3.  The authors need to use Wilcoxon just as they did for metabolomics results. And if results are not significant, then it should be stated that non statistically significant trends were identified (trends are still valid as long as used to guide confirmatory experiments)

Proteomics Data need to be deposited on an academic database such as PRIDE (https://www.ebi.ac.uk/pride/archive/)

The string webtool is not very informative when used as described by authors « PPI analysis of differentially expressed proteins was performed using the publicly 573 available STRING database (https://string-db.org/) [86] setting default parameters and 574 including all active prediction parameters. » Indeed, with default parameters, all types of links are considered (including text mining links). So this is not anymore a protein-protein interaction network. The authors need to either state this matter of fact or use other databases such BioGrid. Since the interactome identified is not fully reliable then The enrichment analysis of the network is not reliable neither. This sould be deleted from the paper. There are other simple tools of enrichment analysis such as EnrichR or TargetMine.  The authors should use such tools and start from the whole set of up-regulated proteins or the whole set of down-regulated proteins (an/or both).

Regarding the validation of results by WB, student’s te test cannot be used. Please use Wilcoxon and if not significant, just state « tendancy toward » and show in supplemental data the 5 WB (n = 5 wright ?)

The same remarks regarding statistical analyses hold true for figure 6

Here are PMID of articles which have not been cited and should be discussed and/or mentionned (at least some of these) :

28139855

28246392

32621747

26666663

24905722

Round 2

Reviewer 2 Report

The authors provided complete and fine answers to the issues I have  raised. This work now suitable for publication

Congratulations!